# Unveiling the Multi-Annotation Process: Examining the Influence of Annotation Quantity and Instance Difficulty on Model Performance

**Pritam Kadasi** and **Mayank Singh**
Department of Computer Science and Engineering
Indian Institute of Technology Gandhinagar
Gujarat, India
{pritam.k, singh.mayank}@iitgn.ac.in

## Abstract

The NLP community has long advocated for the construction of multi-annotator datasets to better capture the nuances of language interpretation, subjectivity, and ambiguity. This paper conducts a retrospective study to show how performance scores can vary when a dataset expands from a single annotation per instance to multiple annotations. We propose a novel multi-annotator simulation process to generate datasets with varying annotation budgets. We show that similar datasets with the same annotation budget can lead to varying performance gains. Our findings challenge the popular belief that models trained on multi-annotation examples always lead to better performance than models trained on single or few-annotation examples.

## 1 Introduction

The process of creating datasets often involves practical constraints such as time, resources, and budget that limit the number of annotators or experts available for collecting annotations (Sheng et al., 2008). As a result, there is a prevalence of single or few labels per instance (depending on the limited number of annotators) in the collected data. However, training models on these datasets pose challenges to their generalization abilities, primarily because the data lacks diversity. With a scarcity of different perspectives and variations in the training data (Basile et al., 2021; Plank, 2022), models may struggle to learn robust representations and fail to generalize effectively (Nie et al., 2020; Meissner et al., 2021).

To address these challenges, the NLP community has highlighted the advantages of utilizing multi-annotator datasets (Davani et al., 2022) and also emphasized the importance of releasing multi-annotator datasets and associated information (cultural and demographic, etc.) (Sap et al., 2022; Hershcovich et al., 2022). However, this approach introduces its own set of challenges. Collecting data with multiple annotators requires significant time, annotation budget, and annotator expertise to ensure the creation of high-quality datasets with diverse perspectives.

Moreover, with a limited annotation budget, it becomes crucial to determine the optimal number of annotators within the given constraints. This not only helps save annotation time and budget but also ensures efficient utilization of available resources. While some research (Wan et al., 2023; Zhang et al., 2021) has provided insights and suggestions on finding the optimal number of annotators, a definitive solution to this problem has yet to be achieved.

Another challenge is the restricted number of annotations available per instance, typically not exceeding 6 – 10, even with a large number of recruited annotators (Plank, 2022). This limitation arises from the considerable annotation efforts required for a large volume of instances. As a result, when models are trained on such datasets, they only capture the opinions and information of a small subset of the annotator pool. Additionally, certain datasets have not released annotator-specific labels or established mappings to individual annotators (Nie et al., 2020; Jigsaw, 2018; Davidson et al., 2017). However, the trend is gradually shifting, and there is a growing recognition that annotator-level labels should be made available (Prabhakaran et al., 2021; Basile et al., 2021; Denton et al., 2021).

This study aims to tackle the challenge of lacking annotator-specific labels by simulating a multi-annotation process. Through this study, we provide insights into how the inclusion of more annotators can introduce variations in model performance and identify the factors that influence this variation. Considering that previous research (Swayamdipta et al., 2020) has highlighted the influence of individual instance difficulty on model performance, we examine how the addition of more annotations alters the difficulty level of instances and conse-

quently affects model performance.

In summary, our main contributions are:

- We propose a novel multi-annotator simulation process to address the issue of missing annotator-specific labels.
- We demonstrate, that increasing the number of annotations per instance does not necessarily result in significant performance gains.
- We also demonstrate, that altering the number of annotations per instance has a noticeable impact on the difficulty of instances as perceived by the model and consequently affects the model performance.

## 2 The Multi-annotated Dataset

In practical scenarios, the annotation process begins by hiring one or more annotators who annotate each instance in the dataset. To enhance the representation of the true label distribution, we have the option to extend this process by recruiting additional annotators. We continue this iterative process until either the annotation budget is exceeded or we observe saturation in the model's performance in predicting the true label distribution. As a result, we obtain multiple annotations assigned to each instance in this multi-annotated dataset.

A multi-annotator dataset $\mathcal{D}$ is formally characterized as a triplet $\mathcal{D} = (X, A, Y)$ in this research paper. The set $X$ represents $N$ text instances, denoted as $x_1, x_2, \ldots, x_N$. The set $A$ corresponds to $M$ annotators, represented as $a_1, a_2, \ldots, a_M$. The annotation matrix $Y$ captures the annotations, with rows indexed by $X$ and columns indexed by $A$. Specifically, $Y = Y[X; A] = Y[x_1, x_2, \ldots, x_N; a_1, a_2, \ldots, a_M]$. In simpler terms, the entry $Y[x_i; a_j]$ stores the label $y_{i,j}$ assigned to instance $x_i$ by annotator $a_j$. Furthermore, an *annotator-set* $A_k$, which comprises $k$ annotators where $1 \leq k \leq M$, is defined. Consequently, the subset of $\mathcal{D}$ restricted to $A_k$ is denoted as $\mathcal{D}_k = (X, A_k, Y')$, where $Y' = Y[X; A_k]$. This paper refers to $\mathcal{D}_k$ as the dataset subset with $k$ annotations per instance. Figure 1 illustrates a toy multi-annotator dataset, showcasing $M$ annotators, and $N$ instances along with its subsets comprising 2 and $k$ annotators.

## 3 Simulating the Multi-annotation Process

Based on our current knowledge, it is worth noting that existing multi-annotator datasets typically

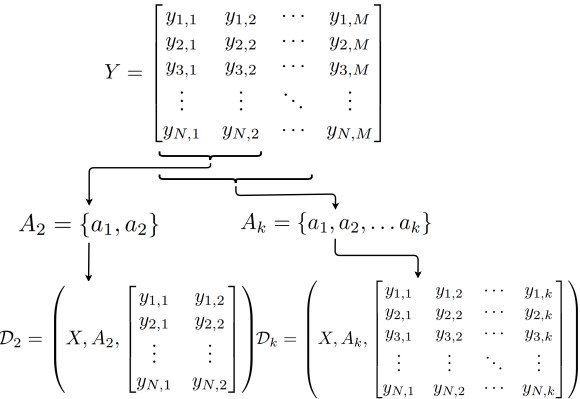

Figure 1: A Toy Multi-Annotator Dataset

do not include annotator-specific labels. Instead, the available information is limited to the label distribution for each instance (Nie et al., 2020; Jigsaw, 2018; Davidson et al., 2017). For instance, in cases with $M$ annotations per instance and three possible labels, the label distribution is commonly represented by a list $[p, q, r]$, where $p$, $q$, and $r$ are positive integers that sum up to $M$. To address this constraint, we introduce a simulation process for multi-annotator scenarios that leverages the instance-level label distribution. Our proposed approach (see Algorithm 1), encompasses the following steps:

- Initially, we generate a list of annotations for each instance by considering the actual instance-level label distribution. [Line 1]
- Subsequently, we randomize these annotation lists using a consistent random seed across instances. [Lines 5–6]
- Next, we select the first $k$ annotations from each randomized list, creating the dataset subset $\mathcal{D}_k$. [Lines 4–8]

By employing this algorithm, we can generate $k$ annotations per instance, thereby addressing the limitation of annotator-specific labels in existing multi-annotator datasets. By repeating the algorithm with different random seeds or parameters, we can create multiple datasets subsets $\mathcal{D}_k$, each containing $k$ annotations per instance. This flexibility enables the generation of diverse subsets, expanding the range of multi-annotator scenarios that can be explored and analyzed in our research.

## 4 Experiments

### 4.1 Datasets

We selected the ChaosNLI dataset (Nie et al., 2020) for our study, as it contains the highest number of

**Algorithm 1** Creation of Annotator Datasets

**Input:** $X$: set of $N$ instances
$\quad\quad\quad\quad$ $CL$: list of C class labels
$\quad\quad\quad\quad$ $LC$: label counts of shape $N \times C$
$\quad\quad\quad\quad$ $M$: number of annotators
**Output:** $\quad$ $\mathcal{D}' = \{\mathcal{D}_1, \mathcal{D}_2, \ldots, \mathcal{D}_M\}$
1: $AL \leftarrow$ GETANNOTATIONLIST()
2: Initialize an empty set $\mathcal{D}'$
3: **for** $k \leftarrow 1$ to $M$ **do**
4: $\quad$ Initialize an empty list $Y'$
5: $\quad$ **for** $i \leftarrow 1$ to $N$ **do**
6: $\quad\quad$ $SL \leftarrow$ RANDOMSHUFFLE($AL[i]$)
7: $\quad\quad$ Choose first $k$ annotations from $AL$
$\quad$ and add it to $Y'$
8: $\quad$ **end for**
9: $\quad$ $\mathcal{D}_k \leftarrow (X, Y')$
10: $\quad$ Add $\mathcal{D}_k$ to $\mathcal{D}'$
11: **end for**
12: **Return** $\mathcal{D}'$

annotations (=100) per instance among the publicly available datasets (Plank, 2022). ChaosNLI is a Natural Language Inference (NLI) task dataset known for its high ambiguity. Additionally, the ChaosNLI dataset includes sub-datasets, namely ChaosNLI-S and ChaosNLI-M, which are subsets extracted from the development sets of SNLI (Bowman et al., 2015) and MNLI-matched(Williams et al., 2018), respectively. Another sub-dataset, ChaosNLI-$\alpha$, is created from the entire development set of AbductiveNLI hereafter, referred to as $\alpha$-NLI (Bhagavatula et al., 2019).

The ChaosNLI dataset consists of 4,645 instances, each annotated with 100 new annotations. Additionally, the dataset already includes 5 old annotations for ChaosNLI-S and ChaosNLI-M, and 1 old annotation for ChaosNLI-$\alpha$. Subsequently, we create $\mathcal{D}_k$'s (see §3) utilizing these datasets and then divide these $\mathcal{D}_k$'s into train, development, and test sets using an 80:10:10 ratio. Table 1 provides detailed statistics of the datasets used in our study.

| Datasets | #Instances | #Annotations Per Instance | #Class Labels |
|---|---|---|---|
| SNLI | 550,152 | 5 | 3 |
| MNLI | 392,702 | 5 | 3 |
| $\alpha$-NLI | 169,654 | 1 | 2 |
| ChaosNLI-S | 1,524 | 100 | 3 |
| ChaosNLI-M | 1,599 | 100 | 3 |
| ChaosNLI-$\alpha$ | 1,532 | 100 | 2 |

Table 1: Dataset Statistics[1]

## 4.2 Pretrained Language Models (PLMs)

In our study, we utilize all the pretrained language models (PLMs) reported in the ChaosNLI work by Nie et al. (2020). Specifically, we experiment with BERT (Devlin et al., 2019), RoBERTa (Liu et al., 2019), XLNet (Yang et al., 2020), ALBERT (Lan et al., 2020), and DistilBERT (Sanh et al., 2020). It is important to clarify that our objective is not to showcase state-of-the-art (SOTA) performance using these models, but rather to demonstrate the variations in performance as we incrementally add annotations to the dataset.

## 4.3 Training Strategies

In this section, we describe two variants of training strategies.
**Majority Label (ML):** The PLMs are finetuned using the majority label, which is determined by aggregating annotations from the target list of annotations. The training objective aims to minimize the cross-entropy between the output probability distribution and the one-hot encoded majority label.
**Label Distribution (LD):** The PLMs are finetuned using the label distribution from the target list of annotations (Meissner et al., 2021). The training objective aims to minimize the cross-entropy between the output probability distribution and the target label distribution.

## 4.4 Evaluation

To evaluate the performance of our models, we utilize the classification accuracy computed on the test dataset. In the ML setting, the accuracy is computed by comparing the label associated with the highest softmax probability predicted by the model with the majority label derived from the target annotations. In the LD setting, the accuracy is computed by comparing the label corresponding to the highest softmax probability predicted by the model with the label that has the highest relative frequency in the target label distribution.

## 4.5 Experimental Settings

Following the approaches described in the studies (Nie et al., 2020; Meissner et al., 2021), we construct base models by finetuning PLMs (described in §4.2) on the combined train sets of SNLI and

---

[1]#Instances corresponding to SNLI, MNLI and $\alpha$-NLI are of train set as only train set is used for training base models in our study.

| Model | Min. Accuracy | | | | | | Max. Accuracy | | | | | |
|---|---|---|---|---|---|---|---|---|---|---|---|---|
| | ChaosNLI-S | | ChaosNLI-M | | ChaosNLI-$\alpha$ | | ChaosNLI-S | | ChaosNLI-M | | ChaosNLI-$\alpha$ | |
| | ML | LD | ML | LD | ML | LD | ML | LD | ML | LD | ML | LD |
| RoBERTa | 0.647 (1) | 0.647 (1) | 0.558 (1) | 0.558 (1) | 0.695 (1) | **0.695 (2)** | 0.75 (100) | **0.741 (20)** | **0.719 (80)** | 0.731 (100) | **0.734 (30)** | 0.73 (30) |
| XLNet | 0.647 (1) | 0.643 (1) | 0.564 (1) | 0.561 (1) | **0.647 (2)** | 0.648 (1) | 0.743 (100) | 0.77 (100) | 0.744 (100) | **0.751 (80)** | 0.695 (100) | **0.685 (30)** |
| ALBERT | 0.639 (1) | 0.639 (1) | 0.568 (1) | 0.568 (1) | 0.668 (1) | 0.668 (1) | 0.796 (100) | 0.737 (100) | 0.706 (100) | **0.751 (90)** | 0.695 (100) | **0.695 (90)** |
| BERT | 0.643 (1) | 0.643 (1) | 0.579 (1) | 0.579 (1) | **0.598 (6)** | **0.585 (6)** | **0.753 (90)** | 0.757 (100) | **0.751 (90)** | 0.769 (100) | **0.613 (3)** | **0.616 (3)** |
| DistilBERT | 0.632 (1) | 0.632 (1) | 0.533 (1) | 0.533 (1) | **0.582 (70)** | **0.584 (90)** | 0.724 (100) | 0.73 (100) | **0.692 (80)** | 0.682 (90) | **0.608 (3)** | **0.61 (3)** |

Table 2: The performance of various models in both the ML and LD settings is presented in this table. Values indicate accuracy, and values in braces indicate $k$. The values highlighted in bold indicate the optimal number of annotators where the performance reaches its peak compared to the maximum annotation budget allocated (100). Conversely, the highlighted values in the minimum accuracy column indicate the lowest performance achieved compared to the minimum budget allocated (1). This information provides insights into the impact of the number of annotators on the model's performance.

MNLI for both ChaosNLI-S and ChaosNLI-M. For the ChaosNLI-$\alpha$ dataset, we construct base models by finetuning on the train set of $\alpha$-NLI. We further finetune these base models with increasing sizes of annotators. Specifically, we finetune models for each $\mathcal{D}_k$, where $k \in [1, 100]$. For each $k$, we report average performance scores over test sets of 10 $\mathcal{D}_k$'s (see §3)

We choose hyperparameters from the experimental settings of the following work (Nie et al., 2020; Meissner et al., 2021; Bhagavatula et al., 2019). Our optimization technique involves employing the AdamW optimizer (Loshchilov and Hutter, 2019). More details on hyperparameters can be found in §A.2. To ensure reproducibility, we conduct our experiments using the open-source Hugging Face Transformers[2] library (Wolf et al., 2020). Furthermore, all experiments are performed using 2 × NVIDIA RTX 2080 Ti GPUs.

## 5 Results and Discussion

### 5.1 Is higher performance always guaranteed by increasing the number of annotations?

Figure 2 presents the accuracy scores as the number of annotations increases. Notably, the trends observed in the performance of ChaosNLI-S, ChaosNLI-M, and ChaosNLI-$\alpha$ challenge the prevailing belief that increased annotations invariably lead to improved performance. Specifically, for ChaosNLI-S and ChaosNLI-M, the accuracy scores exhibit a non-monotonic increasing pattern. In contrast, the trend observed for ChaosNLI-$\alpha$, particularly with BERT and DistilBERT models, deviates from this expected behavior. In these cases, the accuracy scores show a decreasing trend as the number of annotations increases. Upon examining the RoBERTa accuracy scores for the LD setting

in ChaosNLI-S, it is observed that the performance reaches a saturation point between 20 to 80 annotations. This means that increasing the number of annotations beyond this range does not result in significant improvement in the accuracy scores.

Table 2 provides a complementary perspective on the observed trends. It highlights that the minimum performance is not consistently associated with the dataset having the fewest annotations, and vice versa. In the case of ChaosNLI-$\alpha$ with BERT and DistilBERT, it is interesting to note that the optimal performance is achieved with just three annotations. This represents an extreme scenario where a minimal number of annotations can lead to the best performance. In general, these findings shed light on the optimization of our annotation budget. Similarly, the performance gain (maximum - minimum accuracy) across different datasets also significantly varies. The average performance gain for ChaosNLI-M, ChaosNLI-S and ChaosNLI-$\alpha$ is 0.106, 0.177, and 0.031, respectively. The notable variability in performance gain across different datasets further emphasizes that the impact of increasing annotations on performance improvement is not consistent. It underscores the need to carefully analyze and understand the specific characteristics of each dataset and model combination to ascertain the relationship between annotation quantity and performance.

To provide an explanation for the observed complex behavior, we utilize the $\mathcal{V}$-Information (Ethayarajh et al., 2022). $\mathcal{V}$-information is a measure that quantifies the ease with which a model can predict the output based on a given input. The higher the $\mathcal{V}$-information, the easier it is for the model to predict the output given input. Furthermore $\mathcal{V}$-information cannot be negative unless model overfits, etc. (see §A.1).

Figure 3 provides a visual representation of

---
[2] https://huggingface.co/docs/transformers/

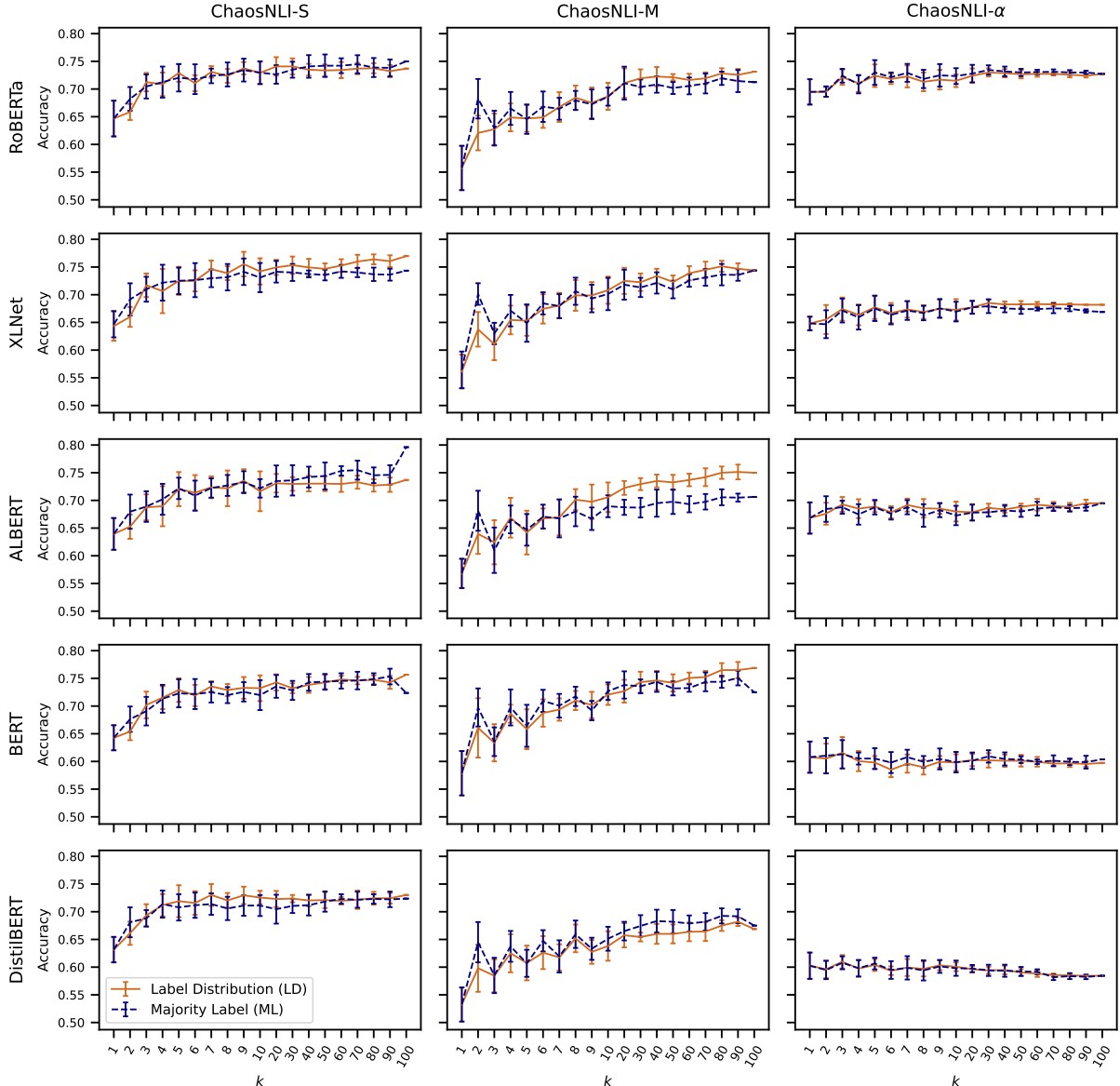

Figure 2: The figure displays accuracy scores for various models across $k$ for datasets ChaosNLI-S, ChaosNLI-M and ChaosNLI-$\alpha$. For every $k$ on X-axis, the mean and standard deviation of the accuracy scores of models trained on 10 $\mathcal{D}_k$'s are displayed. The detailed plots for ChaosNLI-$\alpha$ BERT and ChaosNLI-$\alpha$ DistilBERT can be found in Figure 5 in the Appendix.

the $\mathcal{V}$-information scores for the three datasets across five different PLMs. As anticipated, the $\mathcal{V}$-information scores are higher for the ChaosNLI-S and ChaosNLI-M datasets. Models that exhibit higher $\mathcal{V}$-information scores also tend to yield higher accuracy scores in the LD-based performance evaluation. For instance, RoBERTa outperforms other models (except XLNet, for which the performance is similar) in terms of accuracy for the ChaosNLI-S dataset. The saturation of $\mathcal{V}$-information scores starting at $k = 20$ for the ChaosNLI-S dataset effectively explains the ob-

served saturation of LD-based accuracy after 20 annotations, as depicted in Figure 2. This phenomenon suggests that the model reaches a point where additional annotations provide diminishing returns in terms of extracting valuable insights from the instances. Therefore, the model's performance ceases to improve significantly beyond this threshold. For the ChaosNLI-$\alpha$ dataset, except RoBERTa and XLNet ($\mathcal{V}$-Information $\in [0, 0.25]$, comparatively low), all models yielded approximately zero $\mathcal{V}$-information scores[3]. This implies that adding

---

[3]We used same hyperparameters for all $k$'s due to which

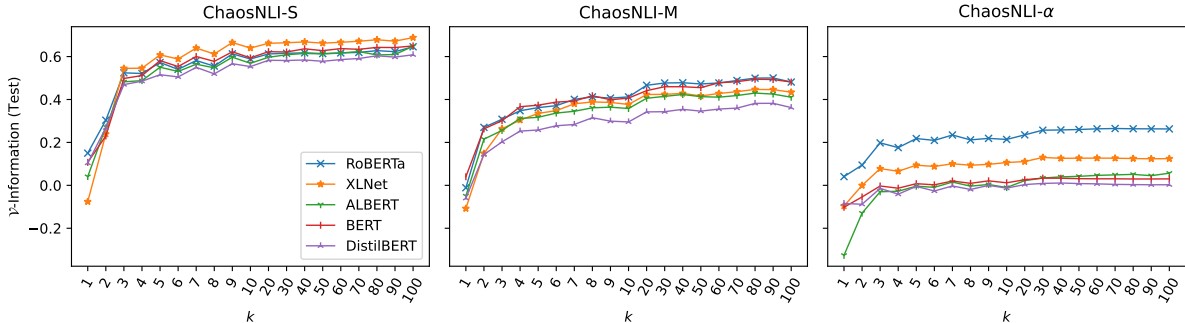

Figure 3: The figure displays the $\mathcal{V}$-Information values for various models in the LD setting. A higher value indicates that the data is easier for the respective model $\mathcal{V}$ with respect to extracting information from it. These values can be compared across datasets and models.

more annotations to the ChaosNLI-$\alpha$ dataset does not establish a clear relationship between the input and output label distribution. This observation suggests that, for this particular variant of the dataset, the model might rely on factors other than the provided annotations to make accurate predictions.

The aforementioned findings indicate that not all datasets yield similar performance when trained under the same budget, underscoring the importance of selecting the appropriate dataset for a specific task. Furthermore, these findings emphasize the significance of determining the optimal number of annotators, as the model's performance varies with the increase in annotations.

### 5.2 Does the number of annotations influence the difficulty of instances as perceived by the model?

To investigate this question, we employ the concept of dataset cartography as proposed by Swayamdipta et al. (2020), which leverages training dynamics to distinguish instances based on their (1) confidence, measured as the mean probability of the correct label across epochs, and (2) variability, represented by the variance of the aforementioned confidence. This analysis generates a dataset map that identifies three distinct regions of difficulty: *easy-to-learn*, *hard-to-learn*, and instances that are *ambiguous* with respect to the trained model. *Easy-to-learn* (**e**) instances exhibit consistently high confidence and low variability, indicating that the model can classify them correctly with confidence. *hard-to-learn* (**h**) instances, on the other hand, have low confidence and low variability, indicating the model's struggle to consistently classify

them correctly over multiple epochs. ***Ambiguous*** (**a**) instances display high variability in predicted probabilities for the true label. We investigate the proportion of the transitions between these categories with the incorporation of additional annotations. For example, **e** → **a** represents proportion of the transitions from *easy-to-learn* to *ambiguous* category among all transitions. This provides valuable insights into the underlying factors that contribute to the observed improvements or lack thereof in the model's performance.

Figure 4 illustrates an interesting pattern in ChaosNLI-S and ChaosNLI-M datasets: as the number of annotations increases, a significant proportion of training instances transition from the **a** → **e** category. For instance, more than 60% of all transitions between 1 to 10 annotations involve instances moving from the **a** → **e** category. However, beyond 10 annotations, the proportion of instances transitioning to the **e** from the **a** category does not show a substantial increase. On the other hand, the reverse transition from the **e** → **a** category is the second most common transition, with an average proportion of 20%. The difference in proportions between the transition from **a** → **e** and the transition from **e** → **a** becomes more substantial (at least 29%) as more annotations are added. In the ChaosNLI-M dataset, we observe a higher proportion of instances transitioning from category **a** to category **h** compared to the ChaosNLI-S dataset. Specifically, over 15% of the ambiguous instances in ChaosNLI-M exhibit a shift towards the hard region, which is more than 50% of similar transitions observed in ChaosNLI-S. We argue that this substantial difference in transition patterns has a direct impact on the performance of models on the ChaosNLI-S dataset compared to ChaosNLI-M.

---

models for $k \leq 3$ overfitted resulting in negative $\mathcal{V}$-Information.

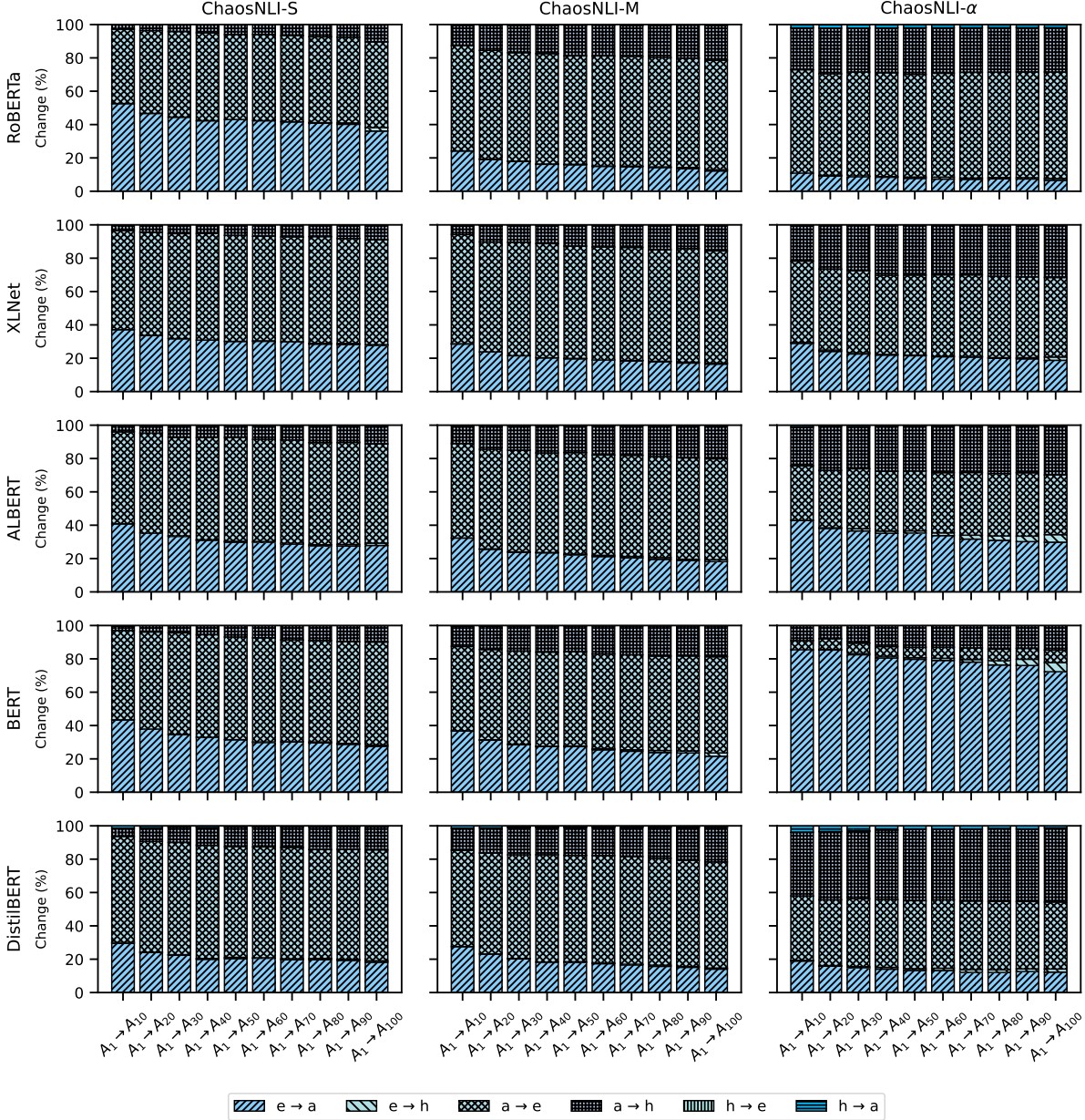

Figure 4: The figure provides a visual representation of the transition of instances between different categories during training as the number of annotators increase from $A_1$ to $A_{10}, \ldots, A_{100}$. $\mathbf{e} \rightarrow \mathbf{a}$ indicates percentage of instances that transitoned from category $\mathbf{e}$ to $\mathbf{a}$.

Despite the presence of higher proportions of $\mathbf{a}$ to $\mathbf{e}$ transitions in ChaosNLI-M compared to ChaosNLI-S, the $\mathbf{a}$ to category $\mathbf{h}$ consistently leads to better performance on the ChaosNLI-S dataset across all models analyzed.

ChaosNLI-$\alpha$ exhibits distinct trends across various models. Specifically, in the case of BERT and DistillBERT, where accuracy scores decline as the annotation increases (see Figure 2), we witness significant proportions of $\mathbf{e} \rightarrow \mathbf{a}$ ($\sim 80\%$) and $\mathbf{a} \rightarrow \mathbf{h}$ ($\sim 43\%$) transitions, respectively. These transitions suggest that the models struggle to comprehend the instances and classify them with reduced confidence. For XLNet and ALBERT, the combined proportion of low confidence transitions, $\mathbf{e} \rightarrow \mathbf{a}$ and $\mathbf{a} \rightarrow \mathbf{h}$ either surpasses or remains equal to the proportion of high confidence transition $\mathbf{a} \rightarrow \mathbf{e}$. In the case of RoBERTa, it behaves the same as ChaosNLI-S and ChaosNLI-M.

These results suggest adding more annotations has indeed its effects on the difficulty of instance thereby affecting the performance of the model.

# 6 Related Works

**Human disagreements in annotations.** Traditional approaches like majority voting or averaging can overlook important nuances in subjective NLP tasks, where human disagreements are prevalent. To address this issue, Multi-annotator models treat annotators' judgments as separate subtasks, capturing the distribution of human opinions, which challenges the validity of models relying on a majority label with the high agreement as ground truth (Davani et al., 2022; Nie et al., 2020). Human variation in labeling, which is often considered noise (Pavlick and Kwiatkowski, 2019), should be acknowledged to optimize and maximize machine learning metrics, as it impacts all stages of the ML pipeline (Plank, 2022). Incorporating annotation instructions that consider instruction bias (Parmar et al., 2023), which leads to the over-representation of similar examples, is crucial. This bias can limit model generalizability and performance. Future data collection efforts should focus on evaluating model outputs against the distribution of collective human opinions to address this issue. All of the above works study annotator disagreements and how they affect the performance of models on downstream tasks. However, in our work, considering disagreements' effect on model performance, we try to find out how the model performance varies as we increase the number of annotations per instance, i.e., varying the annotator disagreement, Overall, we try to answer, does more annotation per instance leads to better performance or is the other way around?

**Annotation under restricted annotation budget.** Also, prior studies have investigated how to achieve optimal performance in natural language processing (NLP) models under restricted annotation budgets. One such study by (Sheng et al., 2008) examined the impact of repeated labeling on the quality of data and model performance when labeling is imperfect and/or costly. Another study by (Bai et al., 2021) framed domain adaptation with a constrained budget as a consumer choice problem and evaluated the utility of different combinations of pretraining and data annotation under varying budget constraints. Another study by (Zhang et al., 2021) explored new annotation distribution schemes, assigning multiple labels per example for a small subset of training examples, and proposed a learning algorithm that efficiently

combines signals from uneven training data. Finally, a study by (Chen et al., 2022) proposed an approach that reserves a fraction of annotations to explicitly clean up highly probable error samples to optimize the annotation process. All these studies contribute to the understanding of how to maximize the performance of NLP models under restricted annotation budgets. Our study aimed to address a specific question within this context: assuming a fixed annotation budget, which dataset would yield the highest performance?

Previous studies have demonstrated that annotation disagreements affect model performance. However, our study aims to explore how performance varies as we change the level of disagreement. we consider ideas from (Zhang et al., 2021) who proposed a learning algorithm that can learn from training examples with different amounts of annotation (5-way, 10-way, 20-way) in a multilabel setting, but we expand the number of annotations from 1-way till 100-way and train our model in a label distribution setting rather than in a multi-label setting. To investigate the reasons for performance variation as we increase the number of annotations, we incorporate (Swayamdipta et al., 2020)'s ideas and (Ethayarajh et al., 2022)'s concepts of dataset difficulty. While previous studies focused on building datasets and models and their impact on performance when the annotation budget is restricted, our work answers whether increasing the annotation budget necessarily leads to improved model performance. Overall, our study aims to demonstrate that, even with less annotation budget than its upper bound, it is possible to achieve optimal performance compared to the performance at the upper bound thereby saving annotation budget and time. Our findings provide insights into optimizing annotation budgets.

# 7 Conclusion

In this paper, we introduced a novel approach to handle the absence of annotator-specific labels in the dataset through a multi-annotator simulation process. Additionally, we investigated the impact of varying the number of annotations per instance on the difficulty of instances and its effect on model performance. Our results highlighted that increasing the number of annotations does not always lead to improved performance, emphasizing the need to determine an optimal number of annotators. This has important implications for optimizing annota-

tion budgets and saving time. Our findings provide valuable insights for optimizing annotation strategies and open up new possibilities for future research in this direction.

## Limitations

The current study acknowledges several limitations that deserve attention. Firstly, the experiments were conducted using small-size Language Models due to resource constraints. It is important to recognize that employing larger language models, such as BLOOM, GPT, and others, could potentially yield different outcomes and should be explored in future research. Furthermore, the scope of the discussion is constrained by the availability of datasets with a large number of labels per instance, leading to the utilization of the ChaosNLI dataset (Nie et al., 2020). Consequently, the generalizability of the findings to other datasets, if they emerge in the future, might be restricted.

## Acknowledgements

We express our gratitude to the anonymous reviewers for their insightful feedback. Our research has received support through the UGC-JRF fellowship from the Ministry of Education, Government of India. Additionally, we would like to extend our thanks to our colleague, Mr. Shrutimoy Das, a Ph.D. student at IIT Gandhinagar, who provided the initial review of this paper and generously shared GPU resources to conduct essential side experiments during critical phases of our research. We are grateful for these contributions, which significantly contributed to the success of this study.

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

Punta Cana, Dominican Republic. Association for Computational Linguistics.

# Appendices

# A  More Details

## A.1  $\mathcal{V}$-Information

$\mathcal{V}$-Information ([Kulmizev and Nivre, 2023](); [Ethayarajh et al., 2022]()), where $\mathcal{V}$ represents specific model families such as BERT, GPT, etc., measures the level of ease with which model $\mathcal{V}$ can predict the output variable $Y$ given the input $X$. The higher the $\mathcal{V}$-Information, the easier it is for the model $\mathcal{V}$ to predict the output variable $Y$ given $X$. To measure $\mathcal{V}$-Information, we use **predictive $\mathcal{V}$-entropy**:

$$H_\mathcal{V}(Y) = \inf_{f \in \mathcal{V}}[-\log_2 f[\varnothing](Y)]$$

and **conditional $\mathcal{V}$-entropy**:

$$H_\mathcal{V}(Y|X) = \inf_{f \in \mathcal{V}}[-\log_2 f[X](Y)]$$

In simple terms, our goal is to find the $f \in \mathcal{V}$ that maximizes the log-likelihood of the label data with and without input $X$. Using these two quantities, $\mathcal{V}$-Information can be calculated using the formula:

$$I_\mathcal{V}(X \to Y) = H_\mathcal{V}(Y) - H_\mathcal{V}(Y|X)$$

It is important to note that $\mathcal{V}$-Information is computed with respect to $H_\mathcal{V}(Y)$, so $I_\mathcal{V}(X \to Y) \geq 0$. Additionally, if $X$ is independent of $Y$, then $I_\mathcal{V}(X \to Y) = 0$.

While $\mathcal{V}$-Information functions as an aggregated measure calculated for the whole dataset, ([Ethayarajh et al., 2022]()) extended this measure to a new measure called Pointwise $\mathcal{V}$-Information (PVI), which allows for the calculation of the difficulty of individual instances. The higher the PVI, the easier the instance is for $\mathcal{V}$ in the given distribution. It can be depicted by the formula:

$$\text{PVI}(x \to y) = -\log_2 p_{f'}(y^*|\varnothing) + \log_2 p_f(y^*|x)$$

where $f_\theta, f'_\theta \in \mathcal{V}$ are models trained with and without input $x \in X$, respectively, and $y^*$ refers to the gold label. Unlike $\mathcal{V}$-Information, PVI can be negative, indicating that the model predicts the majority class better without considering the input $x$ compared to when considering the input.

Refer to Table 6 for a sample of instances from the ChaosNLI-$\alpha$ dataset with very low PVI, which demonstrates the high ambiguity in these instances.

## A.2  Hyperparameter Details

Referring to Table 4, we initially trained the models using the hyperparameters provided by ([Nie et al., 2020]()). However, during our experiments, we observed signs of overfitting to our datasets. Consequently, we adjusted the hyperparameters, leading to the set provided in the table. More hyperparameter details can be found in Tables 3 and 5

## A.3  Detailed Plots for Figure 2

For a more comprehensive view of the phenomenon where performance decreases with an increasing number of annotations, we provide detailed plots for BERT and DistilBERT, as shown in Figure 5. While Figure 2 maintains a consistent y-axis for datasets ChaosNLI-(S, M, and $\alpha$), these plots feature distinct axes.

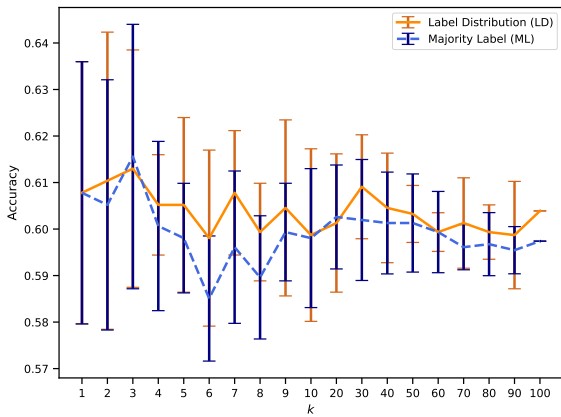

(a) ChaosNLI-$\alpha$ | BERT

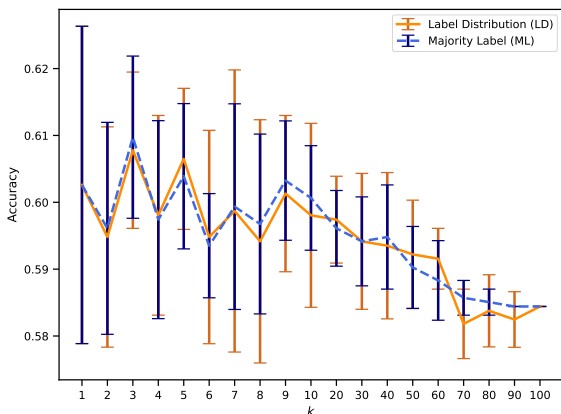

(b) ChaosNLI-$\alpha$ | DistilBERT

Figure 5: The figure displays accuracy scores for BERT and DistilBERT across $k$ for dataset ChaosNLI-$\alpha$. For every $k$ on X-axis, the mean and standard deviation of the accuracy scores of models trained on 10 $\mathcal{D}_k$'s are displayed.

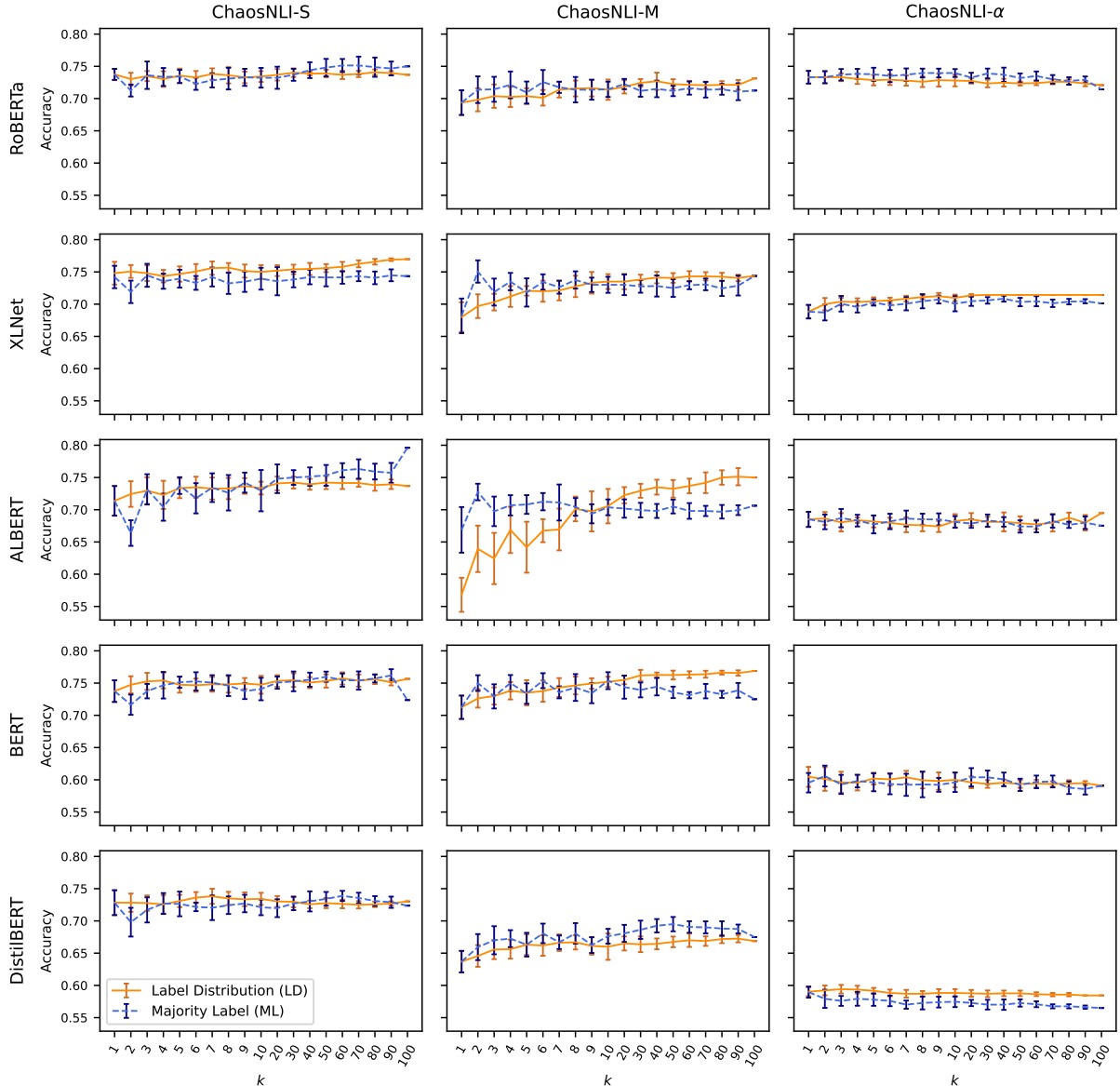

Figure 6: The figure presents accuracy scores for various models across different values of $k$ for datasets ChaosNLI-S, ChaosNLI-M, and ChaosNLI-$\alpha$. Along the X-axis, each $k$ value corresponds to the mean and standard deviation of the accuracy scores, based on models evaluated on test instances with the absolute ground truth.

## A.4   Data Maps

Refer to the RoBERTa datamaps in the LD setting in Figures 7, 8, and 9. For ChaosNLI-$\alpha$, you can find datamaps for BERT and DistilBERT in the LD setting in Figures 10 and 11, respectively.

## B   Results on Absolute Ground Truth

We have extended our evaluation by testing our models on the absolute ground truth, which represents the majority label derived from all 100 annotations. In Figure 6, we provide plots for models trained on datasets with identical training and validation instances as the $\mathcal{D}_k$ datasets. However, the test set remains the same, retaining 100 annotations for the LD setting, where the label distribution of these 100 annotations is considered. In the ML setting, we use the majority label of the 100 annotations.

In Figure 6, on the whole, we observe little to no change in performance as we incrementally increase the number of annotations except few cases. Additionally, it's important to note that the hyperparameters for these models are consistent with those listed in Tables 3, 4 and 5.

| | Parameters | | | | | |
|---|---|---|---|---|---|---|
| **Models** | **Learning Rate** | **Batch Size** | **Weight Decay** | **Max. Epochs** | **Learning Rate Decay** | **Warmup Ratio** |
| **SNLI/MNLI** | 3e-5 | 32 | 0.0 | 3 | Linear | 0.1 |
| **$\alpha$-NLI** | 1e-5 | 8 | 0.0 | 4 | Linear | 0.2 |

Table 3: Hyperparameters for base models RoBERTa, XLNet, ALBERT, BERT and DistilBERT

| **Parameter** | **RoBERTa** | **XLNet** | **ALBERT** | **BERT** | **DistilBERT** |
|---|---|---|---|---|---|
| Learning Rate | 5e-6 | 5e-6 | 5e-6 | 5e-5 | 5e-6 |
| Batch Size | 8 | 8 | 8 | 8 | 8 |
| Weight Decay | 0.0 | 0.0 | 0.0 | 0.0 | 0.0 |
| Max. Epochs | 3 | 5 | 5 | 3 | 3 |
| Learning Rate Decay | Linear | Linear | Linear | Linear | Linear |
| Warmup Ratio | 0.1 | 0.1 | 0.1 | 0.1 | 0.1 |

Table 4: Hyperparameters for finetuned models for dataset ChaosNLI-$\alpha$

| **Parameter** | **RoBERTa** | **XLNet** | **ALBERT** | **BERT** | **DistilBERT** |
|---|---|---|---|---|---|
| Learning Rate | | | 5e-5 | | |
| Batch Size | | | 32 | | |
| Weight Decay | | | 0.0 | | |
| Max. Epochs | | | 3 | | |
| Learning Rate Decay | | | Linear | | |
| Warmup Ratio | | | 0.0 | | |

Table 5: Hyperparameters for finetuned models for dataset ChaosNLI-S and ChaosNLI-M

| Index | Observation 1 | Hypothesis 1 | Hypothesis 2 | Observation 2 | PVI | Current Label | True Label |
|---|---|---|---|---|---|---|---|
| 1 | Jimmy grew up very poor. | A family offered Jimmy to pay his tuition. | He took out a loan for school. | So he repaid them for college. | -5.552362 | 1 | 2 |
| 2 | Jake needed to pick his son up from soccer practice. | Jake forgot and his son had to wait alone for hours at football practice. | Jake left late and got caught in traffic. | His son resented him for it for a long time. | -5.044293 | 1 | 2 |
| 3 | Samuel loved reading old science fiction stories. | He read the Star Wars extended universe material. | Samuel was gifted a science text book. | He loved it! | -4.835824 | 1 | 2 |
| 4 | Lori's class was supposed to be dissecting frogs. | Lori's class didn't take dissection serious. | Lori's teacher confuse a frog on her desk with an instruction booklet. | She picked up a knife and started dissecting the frog. | -4.214444 | 1 | 2 |
| 5 | Lary was a poor coal miner. | Lary enjoyed his job even though he was not good at it. | Larry came across a pile of coal. | Lary was happy and excited. | -4.207170 | 2 | 1 |

Table 6: High ambiguous instances of ChaosNLI-$\alpha$ dataset – RoBERTa - $\mathcal{D}_{100}$ - LD

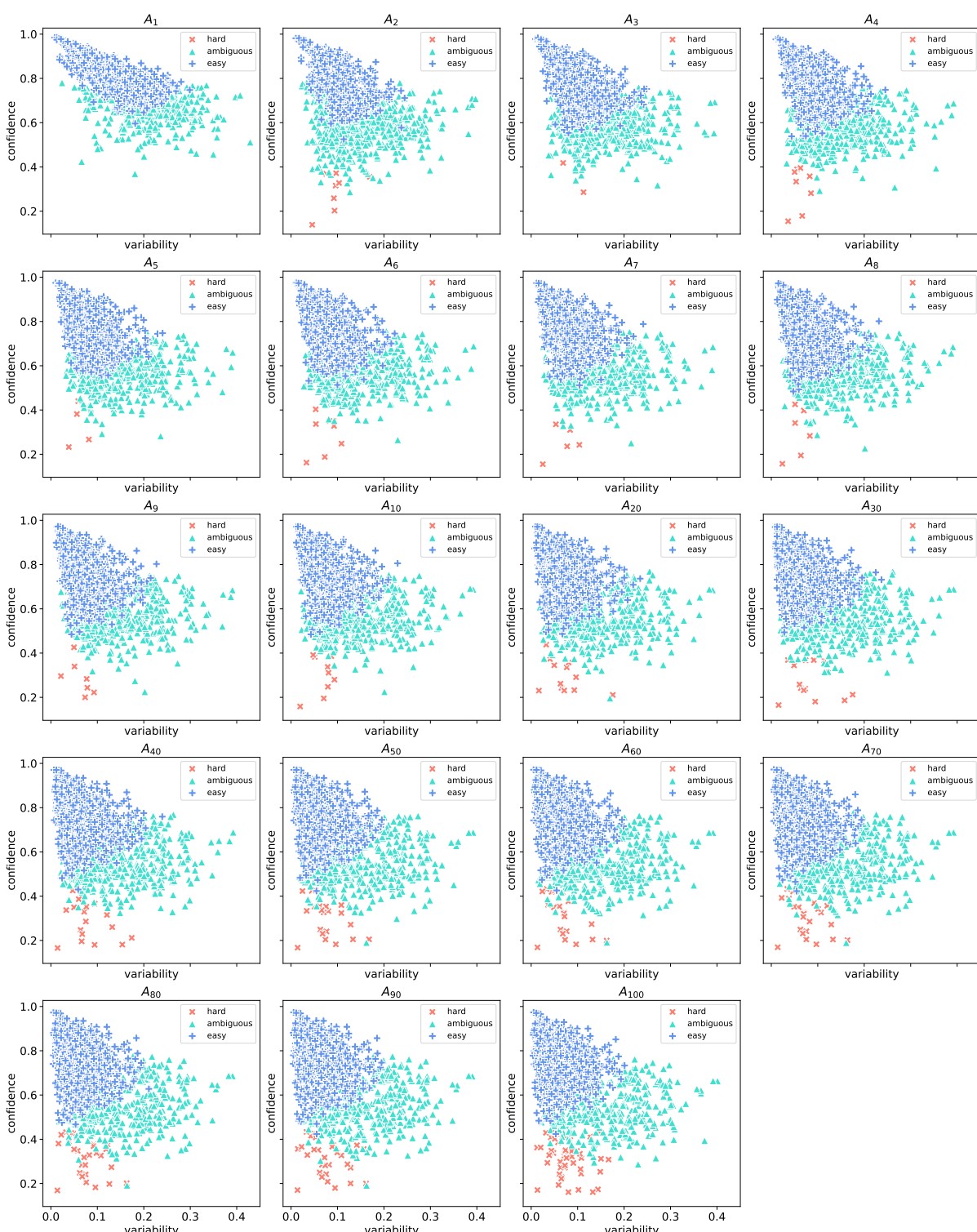

Figure 7: Datamaps across different annotator sets for RoBERTa model trained on ChaosNLI-S dataset in LD setting

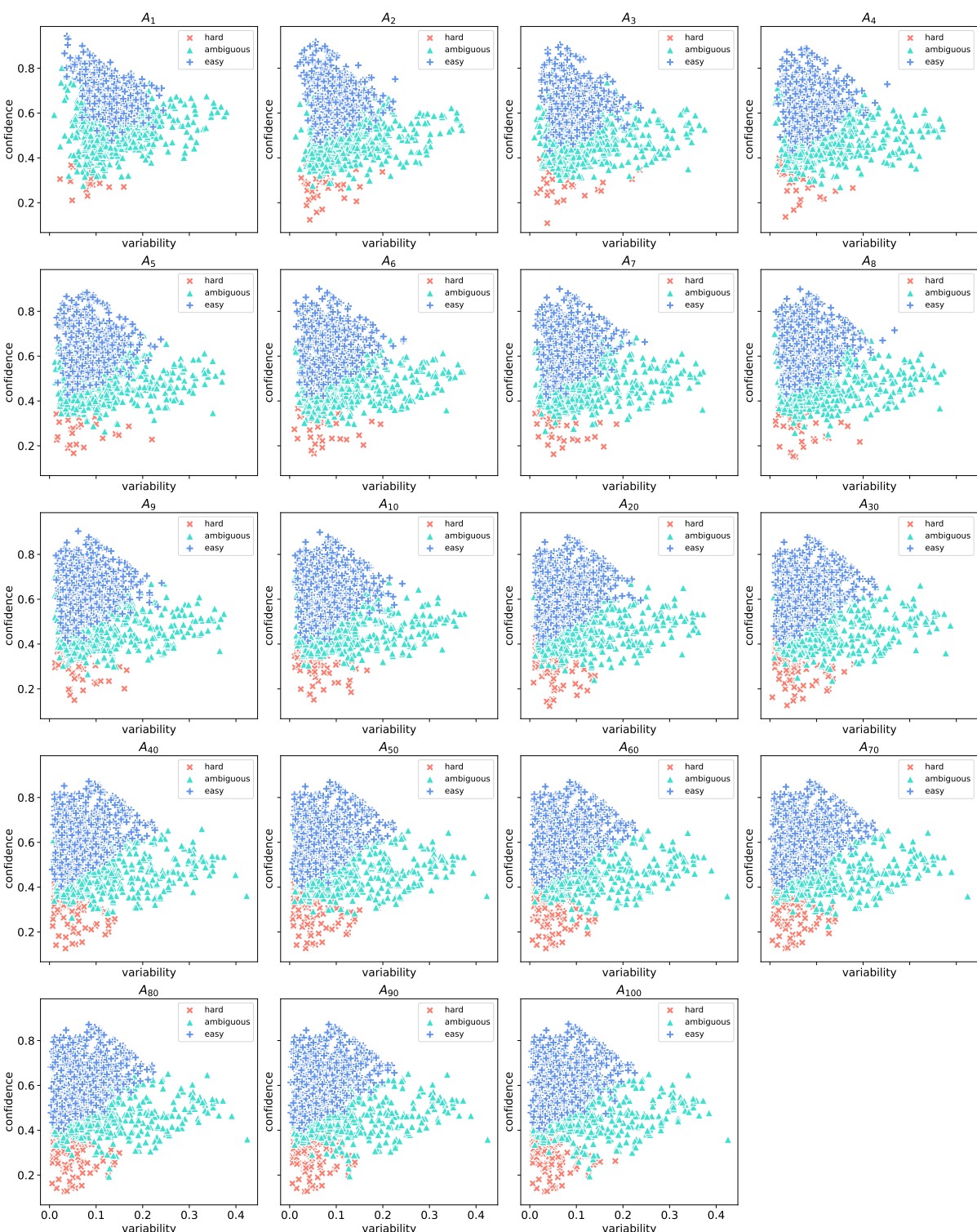

Figure 8: Datamaps across different annotator sets for RoBERTa model trained on ChaosNLI-M dataset in LD setting

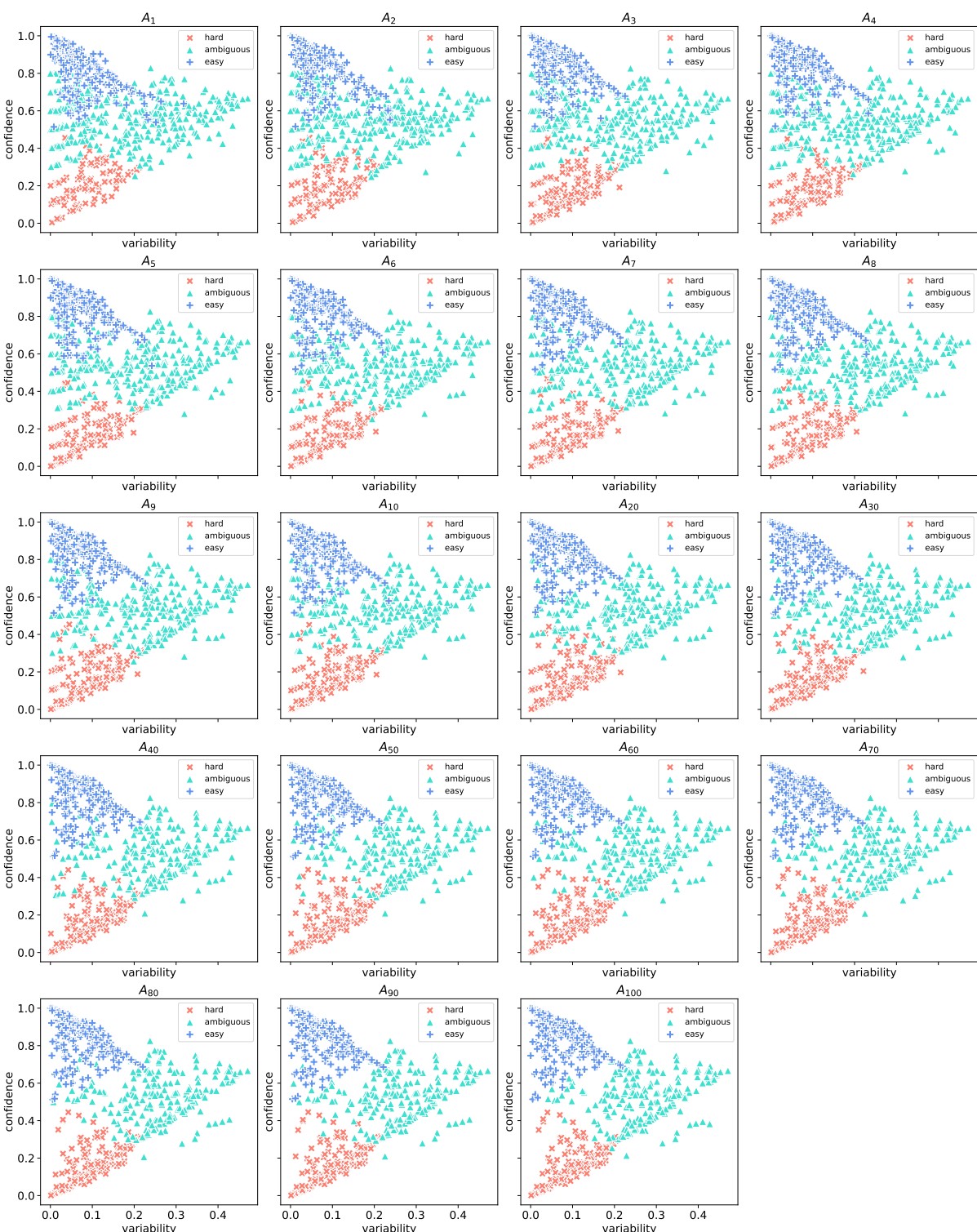

Figure 9: Datamaps across different annotator sets for RoBERTa model trained on ChaosNLI-$\alpha$ dataset in LD setting

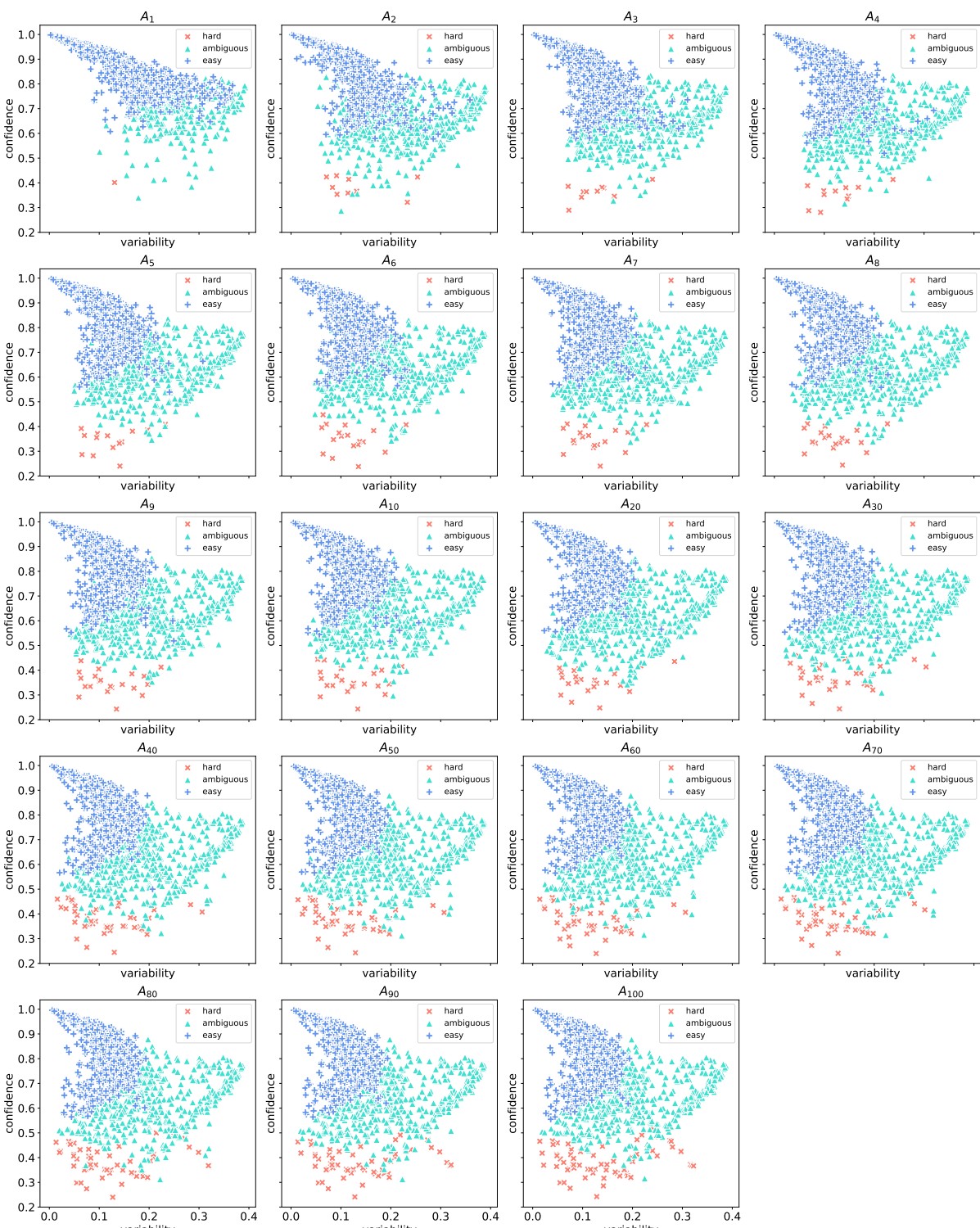

Figure 10: Datamaps across different annotator sets for BERT model trained on ChaosNLI-$\alpha$ dataset in LD setting

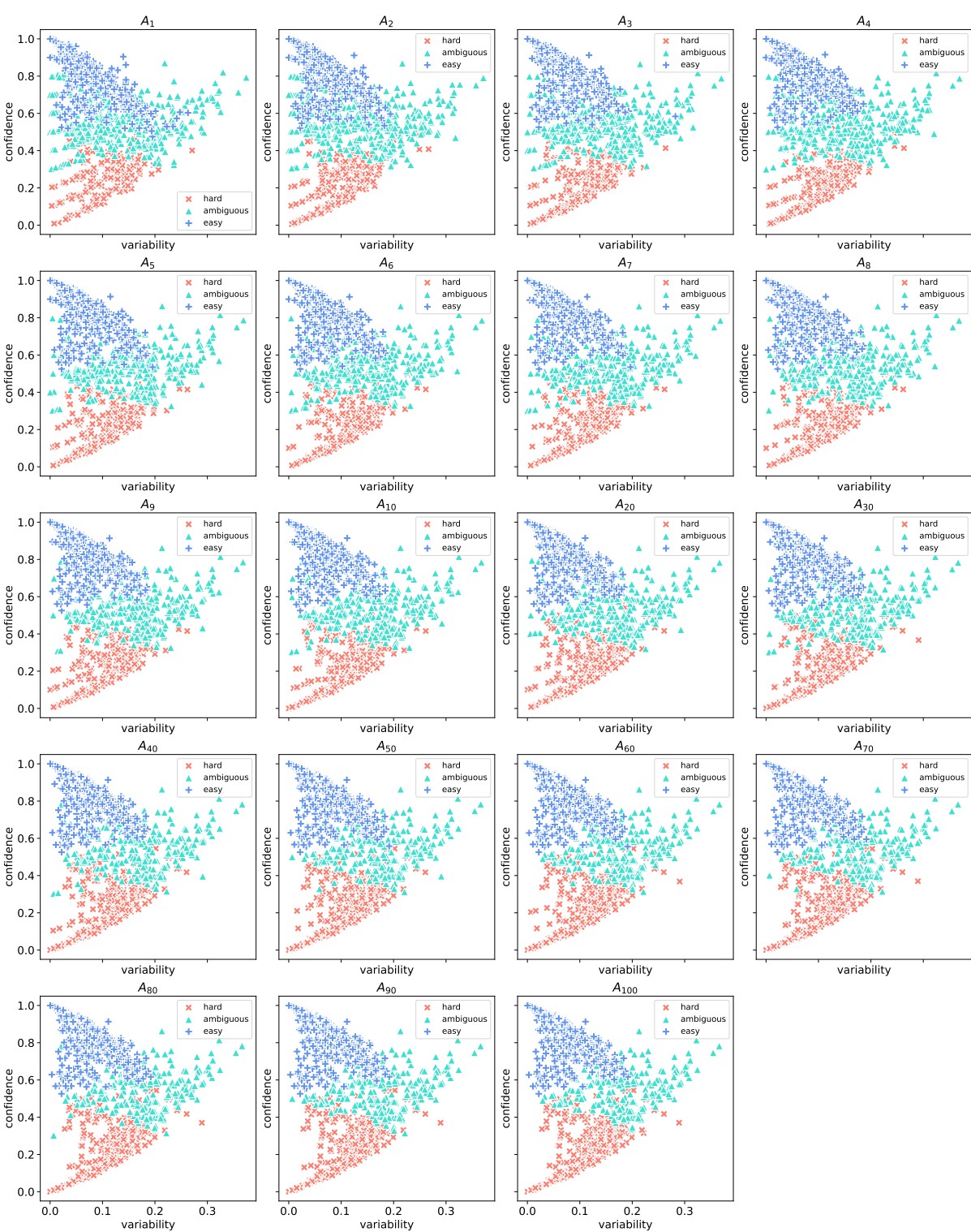

Figure 11: Datamaps across different annotator sets for DistilBERT model trained on ChaosNLI-$\alpha$ dataset in LD setting