# OpenReview forum: "Unveiling the Multi-Annotation Process: Examining the Influence of Annotation Quantity and Instance Difficulty on Model Performance"
_EMNLP/2023/Conference — EMNLP 2023 Findings_

### Official Review · Reviewer_qogc · 2023-08-04

**Soundness:** 3

**Excitement:**

4: Strong: This paper deepens the understanding of some phenomenon or lowers the barriers to an existing research direction.

**Paper Topic And Main Contributions:**

This work studies annotation design choices when collecting new datasets. Specifically, it explores the effect of using a varying number of annotations per instance on the resulting performance of models trained on the collected data. The study is conducted on the textual entailment task using previously collected datasets (ChaosNLI) that contain 100 annotations from different workers per instance.
The authors simulate datasets with K-annotations per instance by sampling K labels from each original instance without replacement.
In the experiments, K varies from 1 to 100, and the annotator set is assumed to be very large.
The results mostly indicate that model performance saturates quickly with the number of annotations per example.

**Reasons To Accept:**

This work tackles an important problem in the collection of new annotated datasets - how to assess the necessary number of answers per prompt to get a useful dataset under a limited budget.
The authors use non-trivial and illuminating analysis methods, such as dataset cartography techniques and other different measures.
The experiments span multiple NLI datasets, several models, and other experimental design choices (e.g. single ground-truth label vs. a gold distribution).
The manuscript is well-organized and easy to follow, even though it deals with many complicated details, it appears that a lot of effort has been put into this work.

**Reasons To Reject:**

I believe that this work did not answer perhaps the most pertinent question that the authors had posed.
Given a fixed budget, a researcher can vary between K - the number of annotators per instance, or N - the number of instances annotated.
While the authors propose increasing K up to some level for better performance, this work doesn't explore what happens to model performance when varying N under the same budget.
The tradeoff between these two parameters is at the heart of many annotation design choices, and in my view, the current contribution of this work is of little use until this question is addressed.
I also suspect that the results in this work are strongly influenced by the small training set in ChaosNLI (~1200 training instances) and that some of these would be invalidated once larger datasets are used (i.e. larger N with the same K). The authors could have easily accomplished that analysis using the original NLI datasets that set K to 5.

Another aspect that I think was overlooked in this study is the overall quality of the dataset attained at different Ks.
The authors should measure the accuracy between the K-dataset and the "absolute" ground truth collected from 100 workers.
When designing a dataset annotation protocol, the correctness of the resulting data is possibly more important than the performance of the model trained on the data.
While a model could achieve good performance at K=3 for example, maybe the K-dataset does not show high inter-annotator agreement. Therefore, maybe the attained quality of the dataset is low and not desirable. Perhaps the analysis in this study should also display model performance against the "absolute" ground truth (K=100) as well.

Another problem in this work is that the title and introductory sections mislead the reader to believe this is a general-purpose study of annotation protocols in NLP, while this work is demonstrated only on textual entailment. Moreover, this work perhaps is more suited for classification tasks, and reproducing similar results on other tasks that require span selection (QA, NER, etc.) or text generation would be much more challenging.

I would also suggest that the authors add a std. dev. to the accuracy results in some of the charts (e.g. chart 2).
I think that the variability in results attained at lower K's would be much more significant compared to higher K's, and that should definitely be also a factor when a researcher designs the annotation protocol.

**Reproducibility:**

4: Could mostly reproduce the results, but there may be some variation because of sample variance or minor variations in their interpretation of the protocol or method.

**Reviewer Confidence:**

4: Quite sure. I tried to check the important points carefully. It's unlikely, though conceivable, that I missed something that should affect my ratings.

---

> ### Author Rebuttal · Authors · 2023-08-29
>
> We wholeheartedly thank you for the invaluable feedback that you have given to this manuscript. We would like to address the concerns raised here.
>
>
> 1. > I believe that this work did not answer perhaps the most pertinent question that the authors had posed. Given a fixed budget, a researcher can vary between K - the number of annotators per instance, or N - the number of instances annotated. While the authors propose increasing K up to some level for better performance, this work doesn't explore what happens to model performance when varying N under the same budget. The tradeoff between these two parameters is at the heart of many annotation design choices, and in my view, the current contribution of this work is of little use until this question is addressed. I also suspect that the results in this work are strongly influenced by the small training set in ChaosNLI (~1200 training instances) and that some of these would be invalidated once larger datasets are used (i.e. larger N with the same K). The authors could have easily accomplished that analysis using the original NLI datasets that set K to 5.
>
>
>    We did not pursue this specific experiment as it has already been addressed by Zhang et al. (2021) (https://aclanthology.org/2021.emnlp-main.601/). whose work has significantly inspired our own research presented in this paper. In their study, they employ a learning objective that enables models to learn from unlabeled, single and multi-label instances. They systematically explore the tradeoff between N and K while maintaining a fixed annotation budget. One of their findings reveal that introducing multi-label examples at the cost of annotating fewer instances yields notable improvements in the NLI task performance. Notably, Zhang et al. (2021) also conducted their analysis on the same ChaosNLI dataset.
>
>    Furthermore, to address concerns related to the dataset size, In our experiments, we establish a base model by finetuning Pretrained Language Models (PLMs) on substantial original training sets, such as SNLI, MNLI, and AbductiveNLI, encompassing over 100K instances (>>1.2K). We further finetune these base models on the ChaosNLI dataset, as elaborated in our paper. Therefore, we believe the concern expressed about the influence of the small training set in ChaosNLI has been effectively mitigated.
>
>    In summary, while we acknowledge the reviewer's important point about varying N under a fixed budget, we believe that this question has been investigated by Zhang et al. (2021), and our work builds upon their insights. Additionally, our experimentation with a comprehensive base model address concerns about the limited training set in ChaosNLI affecting the validity of our findings.
>
> 2.
>     > Another aspect that I think was overlooked in this study is the overall quality of the dataset attained at different Ks. The authors should measure the accuracy between the K-dataset and the "absolute" ground truth collected from 100 workers. When designing a dataset annotation protocol, the correctness of the resulting data is possibly more important than the performance of the model trained on the data. While a model could achieve good performance at K=3 for example, maybe the K-dataset does not show high inter-annotator agreement. Therefore, maybe the attained quality of the dataset is low and not desirable. Perhaps the analysis in this study should also display model performance against the "absolute" ground truth (K=100) as well.
>
>     We have indeed conducted the suggested experiment, but the findings were not reported in the paper due to constraints on page limit. However, we are fully prepared to incorporate these results into the revised version of the paper.
>
>
> 3.  > Another problem in this work is that the title and introductory sections mislead the reader to believe this is a general-purpose study of annotation protocols in NLP, while this work is demonstrated only on textual entailment. Moreover, this work perhaps is more suited for classification tasks, and reproducing similar results on other tasks that require span selection (QA, NER, etc.) or text generation would be much more challenging.
>
>     We acknowledge your observation. The title and introduction may indeed have conveyed a broader scope than our work's specific focus on textual entailment. We recognize that the applicability of our approach and findings primarily pertains to classification tasks. Your feedback is valuable, and we will ensure to address this aspect in the revised version of the paper. Thank you for highlighting this point.
>
> 4.  > I would also suggest that the authors add a std. dev. to the accuracy results in some of the charts (e.g. chart 2). I think that the variability in results attained at lower K's would be much more significant compared to higher K's, and that should definitely be also a factor when a researcher designs the annotation protocol.
>
>     We appreciate your suggestion. We will incorporate the standard deviation (STDDEV) in the accuracy plots in the revised version of the paper. Thank you for bringing this to our attention.

---

### Official Review · Reviewer_KTZY · 2023-08-06

**Soundness:** 3

**Excitement:**

4: Strong: This paper deepens the understanding of some phenomenon or lowers the barriers to an existing research direction.

**Paper Topic And Main Contributions:**


The reported research studies how does an increased number of per-instance annotations impacts the model performance for a specific (although not explicitly mentioned) NLP task and demonstrates that "more" does not always convert into better model performance being the common belief. The research also explores how the number of per-label annotations impacts the "difficulty" of instances and how it affects the model performance.

**Questions For The Authors:**


While it is clear from the results presented in Table 2 that the impact of the increasing number of annotations on the model performance is not consistent across the datasets and models, I was wondering why you did not explicitly mention that at least for ChaosNLI-S and ChaosNLI-M datasets the general trend is in fact that model's performance benefits from more annotations up to a certain number (eg. having around 20 annotations is always better than 2-3)?

**Reasons To Accept:**


- the paper is very clearly written and provides sufficient level of detail to understand the carried-out research

- the work provides interesting new insights into the topic of creation of multi-annotator datasets highly advocated by the community

- the outcome of the study have some practical implications in the context of decisions  related to the creation of corpora vis-a-vis budget envisaged for that purpose

- the authors are well aware of the limitations of the presented results


**Reasons To Reject:**


- I missed the discussion and/or potential references why the proposed simulation of multi-annotator scenario should be correct, it is neither well motivated nor explained

- the reported experiments focus on one specific NLP task and one dataset, which makes the title and the claims made in the paper somewhat misleading

**Reproducibility:**

4: Could mostly reproduce the results, but there may be some variation because of sample variance or minor variations in their interpretation of the protocol or method.

**Reviewer Confidence:**

5: Positive that my evaluation is correct. I read the paper very carefully and I am very familiar with related work.

**Typos Grammar Style And Presentation Improvements:**

- it is not straightforward to understand what is the task behind the ChaosNLI dataset. Clearly, one can read the reference, but for the readability it would be better to adde1-2 sentences about this dataset

- Figure 5 should be mentioned in the main text I believe

- I would suggest to change the title of the paper in order to explicitly reflect the name of the NLP task you dealt with since otherwise
the reader might be mislead and infer that the presented findings are of more generic nature (which is not the case)

---

> ### Author Rebuttal · Authors · 2023-08-29
>
> We wholeheartedly thank you for the feedback and we would like to address the concerns raised.
>
> 1. Referring to the quote below:
>     > I missed the discussion and/or potential references why the proposed simulation of multi-annotator scenario should be correct, it is neither well motivated nor explained.
>
>     The rationale behind this approach is to handle situations where annotator-specific labels are absent. Several datasets, such as (Nie et al., 2020; Jigsaw, 2018; Davidson et al., 2017), lack annotator-specific labels. By introducing the multi-annotator simulation process, we aimed to bridge this gap and enable researchers to understand individual annotator patterns and biases. We encourage the release of annotator-specific labels as this facilitates the model's ability to learn from these individual patterns during training. We will provide a more comprehensive discussion of this motivation and its implications in the updated version of the paper. Thank you for raising this point.
>
> 2.
>     > the reported experiments focus on one specific NLP task and one dataset, which makes the title and the claims made in the paper somewhat misleading.
>
>     We agree, we will address this in the updated version of this paper.
>
> 3.  > While it is clear from the results presented in Table 2 that the impact of the increasing number of annotations on the model performance is not consistent across the datasets and models, **I was wondering why you did not explicitly mention that at least for ChaosNLI-S and ChaosNLI-M datasets the general trend is in fact that model's performance benefits from more annotations up to a certain number (eg. having around 20 annotations is always better than 2-3)?**
>
>     Your insight sheds light on an important aspect, and we appreciate your input. In our initial focus on comparing against the maximum annotation budget of 100, we might have inadvertently overlooked these specific trends. Your suggestion is highly valuable, and we will make sure to integrate this viewpoint in the updated version of the paper. Thank you for bringing this to our attention.
>
> 4. Typos Grammar Style And Presentation Improvements:
>     >it is not straightforward to understand what is the task behind the ChaosNLI dataset. Clearly, one can read the reference, but for the readability it would be better to adde1-2 sentences about this dataset
>
>     Thanks for the suggestion, we will address this in the updated version of the paper.
>
>     >Figure 5 should be mentioned in the main text I believe,
>
>     Yes, we agree, due to page limit, we had to add this figure in the Appendix. We will address this in the revised version of the paper.
>
>     > I would suggest to change the title of the paper in order to explicitly reflect the name of the NLP task you dealt with since otherwise the reader might be mislead and infer that the presented findings are of more generic nature (which is not the case)
>
>     Thanks for the suggestion, our approach and findings are applicable for classification tasks, we will add these changes in the updated version of this paper.

---

### Official Review · Reviewer_hZk1 · 2023-08-06

**Soundness:** 2

**Excitement:**

3: Ambivalent: It has merits (e.g., it reports state-of-the-art results, the idea is nice), but there are key weaknesses (e.g., it describes incremental work), and it can significantly benefit from another round of revision. However, I won't object to accepting it if my co-reviewers champion it.

**Missing References:**

A. Line 087: Hovy et al. 2013 (https://aclanthology.org/N13-1132/) already provide enough information to infer that increasing annotators stops being beneficial after a certain number. This paper discusses how to detect spammers (although they say "MACE does not discard any annotators, but weighs their contributions differently. We are thus not losing information.") and that "if most annotators
answer correctly, majority voting is trivially correct". Therefore, if their model is able to identify the high-quality annotations and having a majority of correct labels makes majority voting trivial, then one can also claim that increasing annotators stops being beneficial after a certain number. In other words, the second contribution statement is too general to be considered novel enough.

**Paper Topic And Main Contributions:**

The authors propose a multi-annotator simulation process which aims to fill in missing gaps in annotated datasets and improve model performance.

**Questions For The Authors:**

A. Line 055: If you're saying, for example, that 5 isn't enough, how many annotations do you think are necessary?

**Reasons To Accept:**

The paper addresses an important issue in annotated datasets, where label breakdowns are not released or certain information relevant to the dataset is not made available.

**Reasons To Reject:**

One of their main contributions of showing how increasing the number of annotators doesn't always lead to better performance can be inferred from another paper 10 years ago (further explanation in Missing References). Additionally, their explanation of the simulation process lacks details that emphasize the novelty of their method.

**Reproducibility:**

3: Could reproduce the results with some difficulty. The settings of parameters are underspecified or subjectively determined; the training/evaluation data are not widely available.

**Reviewer Confidence:**

4: Quite sure. I tried to check the important points carefully. It's unlikely, though conceivable, that I missed something that should affect my ratings.

**Typos Grammar Style And Presentation Improvements:**

A. The abstract should be a little more specific. Can you say a little more about the simulation process you're proposing? What range of annotation budgets are you looking at? What tasks are you training your models for? You want to make sure you give the reader enough information to decide whether they want to read your paper or not (otherwise those who would've been interested may choose not to).

B. Line 092: What do you mean the number of annotations has a noticeable impact on the difficulty of instances? The difficulty of the instance should be inherent to the data itself and shouldn't be affected by who is annotating it.

C. Figure 1 seems to add more confusion than to add clarity. Can you think of a better way to visualize the concept? There are also a lot of new symbols between 106 and 125, which makes it hard to read. Can you show the equations on a different line maybe?

D. I'm not sure the approach makes sense. it just sounds like you're creating a dataset of random labels?

E. Lines 192-193: I would rephrase or remove this. When you clarify that you're not aiming for SOTA performance, it sounds like your paper isn't going to share anything novel.

---

> ### Author Rebuttal · Authors · 2023-08-29
>
> We appreciate your valuable feedback and would like to address the concerns regarding the novelty of our contributions. Regarding the claim that our main contribution about the impact of increasing annotators has been previously addressed, as quoted
>
> >  One of their main contributions of showing how increasing the number of annotators doesn't always lead to better performance has already been published by another paper 10 years ago.
>
> and
>
> > Line 087: Hovy et al. 2013 (https://aclanthology.org/N13-1132/) already show that increasing annotators stops being beneficial after a certain number, so this second contribution can't be considered novel
>
> we would like to address this first:
>
>
> 1. In their 2013 study (MACE), Hovy et al. employed majority label (ML) and MACE as aggregation techniques to establish a singular ground truth. In contrast, our research integrates both the ML and label distribution (LD) approach. The LD method preserves each annotator's individual perspective, unlike ML or MACE. ML disregards minority perspectives, while MACE employs additional annotations to detect and exclude spurious labels. Therefore, our inclusion of LD in experiments aims to encompass all annotator viewpoints. **It is crucial to highlight that the MACE paper addresses a fundamentally distinct problem by assuming a single true label. The context and focus of our experiments markedly deviate from the scope of the MACE paper.** Thus, we refrain from making direct comparisons due to these dissimilarities.
> 2. Furthermore, as we delve into the graphical depictions in the MACE paper (Fig. 6), a clear upward trend in the accuracy curve, becomes apparent. **It is essential to highlight that their analysis is limited to a range of up to 10 annotations. In contrast, our investigation extended to encompass annotation count sufficiently larger than 10, consistently revealing an increasing trend in the accuracy curve.**
>     > Hovy et al. 2013 (https://aclanthology.org/N13-1132/) already show that increasing annotators stops being beneficial after a certain number
>
>     **Nowhere in the MACE paper is there a statement or demonstration that increasing the number of annotators ceases to provide benefits after a certain threshold**. We kindly ask you to identify the specific section or passage in the MACE paper that supports this claim.
>
> 4. Referring to the quote below:
>     > A. Line 055: If you're saying, for example, that 5 isn't enough, how many annotations do you think are necessary?
>
>     The paper offers insights that address this question. Notably, Figure 3 plays a crucial role by illustrating the curve of $\mathcal{V}$-Information values. A significant observation emerges from this depiction: higher $\mathcal{V}$-Information values correspond to increased accuracy. As a result, the identification of the optimal number of annotators can be rooted in pinpointing the peak $\mathcal{V}$-Information value. However, it's important to recognize that arriving at a clear conclusion requires a more thorough investigation.
>
> 5. For the quote below
>     > Line 092: What do you mean the number of annotations has a noticeable impact on the difficulty of instances? The difficulty of the instance should be inherent to the data itself and shouldn't be affected by who is annotating it.
>
>     > **"The difficulty of the instance should be inherent to the data"**
>
>     Certainly, We will rephrase that part as: "We also demonstrate, altering the number of annotations per instance has a noticeable impact on the difficulty of instances **as perceived by the model** and consequently affects the model performance." Thanks for highlighting this point
>
> 6. Typos Grammar Style And Presentation Improvements (A & C): Thanks for the suggestion, we will add these changes to the updated version of this paper.

---

### Meta-Review · Area_Chair_88tP · 2023-09-05

**Recommendation:** 4
**Best Paper Recommendation:** No

**Metareview:**

The paper examines the impact of number of annotators per instance on model performance for text classification. R2 and R3 see the topic of the paper as important and that the results have practical implications. Additionally, R3 sees the experimental setup as thorough for the particular tasks, datasets, and models used. However R1 indicates that one of their primary results may have already been demonstrated before, limiting the degree to which the paper makes a contribution to the literature on this topic. The authors address this in their rebuttal – there are some similarities in the finding but the scale of experimentation and the nature of the datasets used in this work differ sufficiently. The scope may potentially be limited, as the paper only looks at NLI, and R3 additionally describes some potential issues with the utility and generality of the findings. The discussion helped to clarify some of the concerns of the reviewers; while the experiments may be specific to the one dataset tested (ChaosNLI) the paper does give insight into the impact of annotator volume for ambiguous labeling tasks.

**Meta-Review:**

The paper examines the impact of number of annotators per instance on model performance for text classification. R2 and R3 see the topic of the paper as important and that the results have practical implications. Additionally, R3 sees the experimental setup as thorough for the particular tasks, datasets, and models used. However R1 indicates that one of their primary results may have already been demonstrated before, limiting the degree to which the paper makes a contribution to the literature on this topic. The authors address this in their rebuttal – there are some similarities in the finding but the scale of experimentation and the nature of the datasets used in this work differ sufficiently. The scope may potentially be limited, as the paper only looks at NLI, and  R3 additionally describes some potential issues with the utility and generality of the findings. The discussion helped to clarify some of the concerns of the reviewers; while the experiments may be specific to the one dataset tested (ChaosNLI) the paper does give insight into the impact of annotator volume for ambiguous labeling tasks.

---

### Decision · Program_Chairs · 2023-10-07

**Decision:**

Accept-Findings

**Comment:**

The paper examines the impact of number of annotators per instance on model performance for text classification. R2 and R3 see the topic of the paper as important and that the results have practical implications. Additionally, R3 sees the experimental setup as thorough for the particular tasks, datasets, and models used. However R1 indicates that one of their primary results may have already been demonstrated before, limiting the degree to which the paper makes a contribution to the literature on this topic. The authors address this in their rebuttal – there are some similarities in the finding but the scale of experimentation and the nature of the datasets used in this work differ sufficiently. The scope may potentially be limited, as the paper only looks at NLI, and R3 additionally describes some potential issues with the utility and generality of the findings. The discussion helped to clarify some of the concerns of the reviewers; while the experiments may be specific to the one dataset tested (ChaosNLI) the paper does give insight into the impact of annotator volume for ambiguous labeling tasks.